# Fabricating Porous Carbon Materials by One-Step Hydrothermal Carbonization of Glucose

Ziyun Yao, Wenqi Zhang * and Xinying Yu

College of Chemistry and Chemical Engineering, Shanghai University of Engineering Science, 333 Longteng Road, Shanghai 201620, China; 13956886716@163.com (Z.Y.); 15082461771@163.com (X.Y.)
* Correspondence: zhangwenqi@sues.edu.cn; Tel.: +86-159-0074-6789

**Abstract:** The present study concerned the production of glucose-based porous carbon materials by a one-step acid-catalyzed HTC. The samples were characterized by elemental analysis (EA), scanning electron microscope (SEM), Brunauer–Emmett–Teller (BET), Fourier transform infrared (FTIR) and point of zero charge (pzc). Experimental results showed that the addition of sulfuric acid (SA) with different dosages in the HTC system could improve the yield of products and reduce chemical oxygen demand (COD) of the process water. When the glucose and acid was at a mass ratio of 1:4 (glucose: SA = 1:4), the hydrochar obtained (H-G9) had a larger specific surface area ($S_{BET}$ = 296.71 m$^2$/g) and higher abundance of functional groups on the surface than that of other samples, such as sulfur-containing functional groups and carboxylic groups, belonged to the mesoporous material with highly negatively surface charged. H-G9 exhibited the optimum adsorption for methylene blue (MB). H-G9 adsorbed MB with an initial concentration of 10 mg/L at pH 6 and 25 °C. The adsorption isotherm of MB on H-G9 demonstrated that Freundlich isotherm could be better applied. Regeneration efficiency of 88% was achieved by HTC process for saturated H-G9. This study prepared a porous carbon material by the simple one-step hydrothermal carbonization of glucose in the presence of SA. The maximum monolayer adsorption capacity as high as 332.46 mg/g for MB, which was well beyond that of commercial activated carbon (259.37 mg/g). This indicates that H-G9 has great potential for the removal of MB from wastewater.

**Keywords:** hydrothermal carbonization; glucose; sulfuric acid; porous carbon materials; adsorption

## 1. Introduction

Biomass materials have the advantages of being available in large quantities, being inexpensive and sustainability [1], which occupy an important place in the field of energy and functional carbon material preparation [2]. Many studies showed that hydrothermal carbonization (HTC) is a sustainable way for conversion of biomass materials because of its low operating temperature [3], no pretreatment of dehydration and simple equipment [4].

Many researchers used carbohydrate hydrolysates-monosaccharides (e.g., glucose) as the biomass model compound to study the reaction conditions [5,6] and the transformation mechanism of biomass in HTC process [7,8]. Glucose can be converted into carbon-rich microspheres through mild processing conditions (typically producing well-dispersed, micrometersized carbon spheres); for example, Yao et al. [9] investigated the formation of carbon spheres (contains a dense hydrophobic carbon core) prepared from glucose followed by carbonization at low temperatures (170–180 °C). Mi et al. [10] synthesized carbon micro-spheres by a glucose hydrothermal method.

However, the HTC process could only convert glucose and other biomass into carbon-rich brown microspheres with small pore volume and limited surface area [11], and the activation process was commonly used to prepare porous carbon material similar to activated carbon. For example, Lin's team synthesized activated carbon from glucose using HTC (170–280 °C) and KOH activation (500–700 °C at 1 h) [12]. Ren's group developed

N and S co-doped porous carbon spheres with good adsorption property and reusability, which was prepared by a two-step process (HTC and subsequent chemical activation). Thus, the high-temperature activation results in huge energy consumption and large loss of functional groups. It will be a valuable work to directly prepare porous carbon materials by one-step hydrothermal method.

Several researches explored the effects of different pH and different kinds of acids on the hydrothermal carbonization of glucose [13,14], and they found that the rate of glucose dehydration could be highly promoted [15], and the yield and chemical properties of hydrochar were changed greatly under acidic conditions. However, hydrochars from these studies were difficult to be used as adsorbents directly due to their low porosity.

In the previous work, phenol-containing and nonylphenol ethoxylate (NP-10) waste liquid were adopted as hydrothermal carbonization precursors, respectively [16,17], porous carbon materials with high surface area and porosity were successfully prepared by adjusting the amount of acid. On this basis, this study concerned the production of glucose-based porous carbonaceous material by a one-step process in the presence of acid, and will provide a new route for the future conversion of biomass.

## 2. Materials and Methods

### 2.1. Materials

Sulfuric acid (SA) (AR, 98%) and activated carbon (AC) (AR, 200 mesh) were purchased from Shanghai Titan Scientific Co., Ltd. (Shanghai, China). Glucose (Glu) (AR, 99%), methylene blue ($C_{16}H_{18}ClN_3S \cdot 3H_2O$), 0.01 N hydrochloric acid (HCl) and 0.01 N sodium hydroxide (NaOH) were purchased from Sinopharm Chemical Reagent Co., Ltd. (Shanghai, China). Deionized water was produced by ultrapure water systems (Heal Force, Shanghai, China).

### 2.2. HTC Experiments

Glucose was used as precursor, the mixture of 4 g glucose, 6 mL deionized water and different amounts of SA (0–9 mL) were placed in 50 mL Teflon-lined stainless steel autoclave (Binhai, Jiangsu, China) and were kept at 180 °C for 4 h. The conditions of reaction temperature and time were determined from previous studies [18,19].

At the end of reaction, the reactor was cooled down to room temperature. The liquid-solid mixture was separated by vacuum filtration through filter paper, the solid product was recovered, rinsed with 200 mL deionized water, and then dried in an oven at 80 °C for 12 h until its weight was stable. The obtained hydrochars were denoted as H-GX, where X was the dosage of SA. For instance, H-G0, H-G1, H-G3, H-G6 and H-G9, respectively, adding 0, 1, 3, 6 and 9 mL SA to the glucose solution, the mass ratios Glu: SA = 4:0, 2:1 2:3, 1:3 and 1:4. Additionally, the experiments were all repeated three times.

Hydrochar yield (Y) was calculated using the following Equation (1). The analytical procedures for chemical oxygen demand (COD) were performed according to standard methods [16,17].

$$Y(\%) = \frac{\text{Mass of dry hydrochar}}{\text{Mass of dry glucose}} \times 100\% \tag{1}$$

### 2.3. Characterizations of Hydrochar

The elemental compositions of the hydrochars were determined using elemental analyzer (EA) (vario EL cube CHNOS, Berlin, Germany). Zeta potential values were recorded using a zeta potential analyzer (Zetasizer Nano ZS90), the point of zero charge (pzc) of hydrochars was calculated from the zeta potentials in various pH environments. The chemical structure of the solid product was investigated by FTIR (Nicolet AVATAR 370, Madison, WI, USA), samples were scanned from 4000 $cm^{-1}$ to 400 $cm^{-1}$ with a resolution of 4 $cm^{-1}$. The surface morphological characteristics were characterized using a scanning electron microscope (SEM) (HITACHI S–3400, Tokyo, Japan). Nitrogen adsorption/desorption measurements were analyzed by using a gas sorption analyzer (Micromeritics APSP 2460,

Norcross, GA, USA), the specific surface area was calculated using the Brunauere–Emmette–Teller (BET) model.

### 2.4. Adsorption Experiment

The effects of operating conditions (pH, adsorption time, initial dye concentration, and adsorption temperature) on MB removal were studied. In all experiments, the adsorbents obtained were ground and screened by 200 mesh sieves. Kinetic adsorption experiments were performed at 25 °C, pH = 6.5. 15 mg of the adsorbent was added to a MB aqueous solution (400 mL, 10 mg/L) with a stirring speed of 150 r/min. Samples were taken at certain interval time, filtered through a 0.22 μm membrane and analyzed using an ultraviolet-visible (UV-Vis) spectrophotometer at 664 nm (UV-5100, Shanghai, China). Adsorption experiments were repeated three times. The adsorption capacity at time t ($Q_t$, mg/g) and at equilibrium ($Q_e$, mg/g) were calculated using Equations (2) and (3):

$$Q_t = [(C_0 - C_t)/M] \times V \tag{2}$$

$$Q_e = [(C_o - C_e)/M] \times V \tag{3}$$

$C_0$, $C_t$ and $C_e$ are the initial, time t and equilibrium concentration of pollutants in solution, respectively. M indicates the mass of the adsorbent (g) and V represents the volume of the solution (L).

The pseudo-first-order and pseudo-second-order kinetic models were adopted to analyzed adsorption kinetics data. The models are shown as Equations (4) and (5):

$$\ln(Q_e - Q_t) = \ln Q_e - k_1 t \tag{4}$$

$$t/Q_t = 1/\left(k_2 Q_e^2\right) + t/Q_e \tag{5}$$

where $k_1$ (min$^{-1}$) and $k_2$ (g mg$^{-1}$ min$^{-1}$) are, respectively, the rate constants of pseudo-first order and pseudo-second order kinetic models, t is the contact time(min).

The adsorption isotherm was conducted by adding 15 mg adsorbent to 400 mL solutions with initial MB concentrations of 10, 15, 20, 25, 30, 35, 40 and 45 mg/L, respectively. The adsorption isotherms of MB on hydrochars at 25, 35, 45 and 55 °C were explored in experiments. The supernatant was taken to determine the $C_e$ when the adsorption equilibrium was reached. The adsorption isotherms data were calculated by Langmuir and Freundlich models. The models are shown as Equations (6) and (7):

$$Q_e = \frac{Q_m K_L C_e}{1 + K_L C_e} \tag{6}$$

$$Q_e = K_F \times C_e^{1/n} \tag{7}$$

$Q_m$ (mg/g) is the maximum adsorption capacity, $K_L$ (L/mg) is the constant. $K_F$ (mg/g) is an indicator of adsorption capacity and n is the parameter of average site energy.

To compare the adsorption capacities under acidic, neutral and alkaline conditions, the adsorption experiments were carried out at pH 2, 4, 6, 8 and 10. The pH of the dye solution was maintained by using 0.01 N HCl and 0.01 N NaOH solutions. The equilibrium adsorption capacity, $Q_e$ (mg/g) was calculated by using Equation (3).

### 2.5. Adsorbent Regeneration Experiment

To evaluate the recovering ability of adsorbents, the material prepared for adsorption of MB was recovered simply by filtering with filter paper, washing with deionized water several times and drying at 105 °C. Then, the material was used as precursor, the mixture of the material, deionized water and SA were placed in 50 mL Teflon-lined stainless steel

autoclave (Binhai, Jiangsu, China), and kept at 180 °C for 4 h. At the end of reaction, the reactor was cooled down to room temperature. The liquid–solid mixture was separated by vacuum filtration through filter paper, the solid product was recovered, rinsed with 200 mL deionized water and then dried in an oven at 80 °C for 12 h until its weight was stable. The new adsorbents obtained were ground and screened by 200 mesh sieves. Adsorption experiment by adding 80 mg new adsorbent to a MB aqueous solution (500 mL, 80 mg/L) with a stirring speed of 150 r/min. The adsorbent was repeated several times for adsorption experiments.

## 3. Results and Discussion

### 3.1. Hydrochar Yields and Distribution of Organic Carbon between Hydrochar and Process Water

The hydrochar (H-G0) obtained without the use of SA was a brown substance like humins and the yield was only 25.81 wt% (Figure 1), indicating that the degree of glucose carbonization was low. As the dosage of SA increased from 1 mL to 6 mL (Glu: SA from 2:1 to 1:3), the hydrochar yield increased rapidly. When the dosage of SA was higher than 6 mL, the yields stayed relatively constant and the color of products were black. Firstly, the acid catalyzed hydrolysis of glucose macromolecules accelerated and dehydrated them into small molecules (5-HMF), and then, the concentration of small molecules in the solution increased. As a result, the degree of carbonization of hydrochar under acid-catalyzed conditions was high.

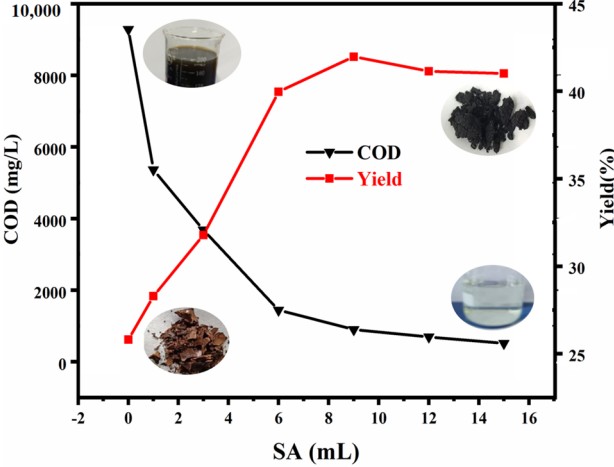

**Figure 1.** Yield of hydrochars and COD of process water changes for different dosage of SA.

It was also seen in this experiment that the color of the process water changed from cloudy brown to clear yellow and further became nearly colorless with the increase in acidity. This might be due to the degree of aromatization was improved. Firstly, when no acid was added, the color of the process water was cloudy brown, which was presumed to be some reaction intermediate (water-soluble monomers and oligomers) based on previous studies [5], the color of the process water then became clearer after the acid, indicating that the intermediate disappeared and further aromatization and polymerization reactions occurred to form carbon spheres [7,15]. The COD of process water also decreased greatly as the dosage of SA increased from 1 mL to 6 mL (Glu: SA from 2:1 to 1:3), corresponding to the improvement of hydrochar yield, and suggested that more organic carbon in glucose solution convert into solid products. When the dosage of SA increased from 6 to 9 mL, the COD of process water continued to decrease from 1440 mg/L to 512 mg/L.

The results of elemental contents of C, H, O and S are summarized in Table 1. The C content of the sample (H-G0) was higher than that reported in other literature, but the O content was lower [12]. According to the Van Krevelen diagram, the phenomenon of O/C atomic ratios decreasing and the H/C basically unchanging was perhaps due to the

improving of glucose decarbonylation by the high concentration substrate used in this experiment [20].

**Table 1.** Elemental composition of hydrochars.

| Sample | Sulfuric Acid (mL) | Elemental Composition (wt%) | | | | | |
|---|---|---|---|---|---|---|---|
| | | C | H | O | S | O/C | H/C |
| H-G0 | 0 | 63.15 | 4.58 | 31.82 | -- | 0.50 | 0.07 |
| H-G1 | 1 | 66.56 | 4.90 | 28.22 | 0.15 | 0.42 | 0.07 |
| H-G3 | 3 | 66.77 | 5.07 | 27.44 | 0.54 | 0.41 | 0.08 |
| H-G6 | 6 | 63.94 | 4.75 | 29.82 | 1.27 | 0.46 | 0.07 |
| H-G9 | 9 | 60.93 | 4.08 | 31.98 | 2.85 | 0.52 | 0.07 |

It is the general law for biomass that HTC process will cause an increase in C content and decrease in O content [21,22]. The characteristics of H-G1 and H-G3 obtained in the condition of adding small amount of SA were consistent with this law, because the acid played a role in accelerating the rate of glucose hydrolysis and favored the carbonization process [23–25].

However, when the dosage of SA was further added (6–9 mL), the C content decreased and the O/C atomic ratios increased, which was possible owing to the degree of oxidation that increased and more oxygen-containing groups occurred on the surface of hydrochars [26]. Meanwhile, the content of S element increased, which may be caused by the presence of sulfur-containing functional groups (e.g., -SO$_3$H) [27].

*3.2. Hydrothermal Carbon Morphologies*

Figure 2 shows the morphologies of the hydrochars prepared with/without SA additives. In the absence of acid, two batches aggregates made of spherical particles with different size could be observed in Figure 2a. The diameter of first batch spheres was in the range of 2–5 μm, apparently, which was larger than that reported in the literature [10]. This might be attributed to higher concentration of the glucose solution used in the experiment [28,29]. The second batch spheres with diameter of around 200 nm were smaller than that of the first batch, which may be produced at later growth stage with the decreasing glucose concentration [30]. Typically, HTC of glucose with low concentration tends to produce monodisperse carbon spheres, which are about 150 nm in diameter [31]. As an example, the uniform microspheres produced from 4.5 wt% glucose solution with an overall size of 150 nm [32], glucose was subjected to hydrothermal carbonization in aqueous solution (25 wt%) at 180 °C for 4 h and resulted carbon nanospheres with diameters of 100 to 200 nm [33].

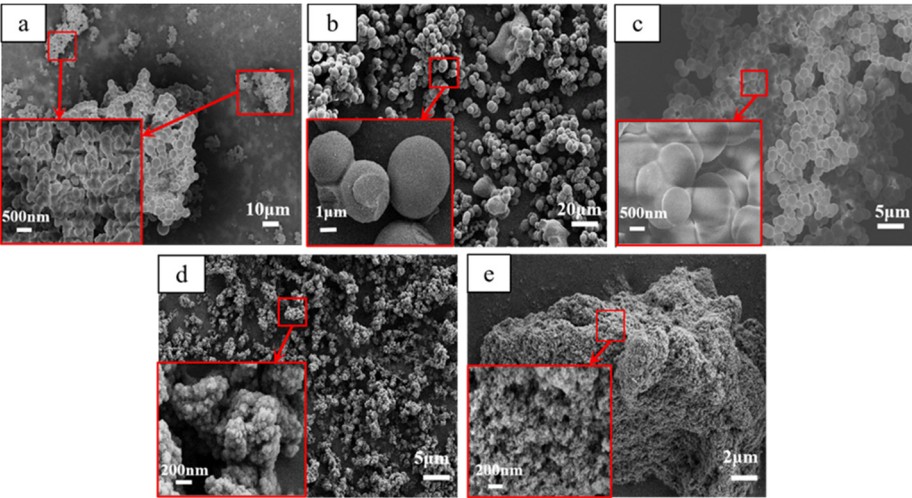

**Figure 2.** SEM micrographs of (**a**) H-G0, (**b**) H-G1, (**c**) H-G3, (**d**) H-G6, (**e**) H-G9.

When 1 mL SA was added, only one batch of aggregates of carbon spheres in size range of 2–5 μm could be observed (Figure 2b). This was possibly due to the increase in the carbon nucleation and growth rate in the acid condition and, thus, there was no obvious batch difference in the reaction process [34]. With increasing the amount of SA from 1 to 3 mL (Figure 2c), the particle size of the carbon spheres decreased to 1–2 μm and agglomeration became heavier, which may have been due to the increase in nucleation rate. Figure 2d shows that as the SA increased to 6 mL, the particle sizes further decreased to 20–150 nm, and these particles agglomerated together to form clusters, resulting in the emergence of pore structures. As the amount of SA to 9 mL (Glu: SA = 1:4) further increased, larger aggregates that were accumulated by nanoparticles with diameters of 10 nm appeared and the particles had no obvious spherical shape (Figure 2e). The porous structure accumulated by nanoparticles for H-G9 further developed due to its decreased particle size.

### 3.3. The Textural Properties of Hydrothermal Carbon

Nitrogen adsorption/desorption isotherms were measured to determine the specific surface area, pore size, and pore volume of hydrochars prepared in different dosages of SA. As shown in Figure 3a, the isotherms of the H-G0, H-G1, H-G3 and H-G6 exhibited a typical II-type curve [35], indicating the pores of these hydrochars were not developed or even non-porous. However, the adsorption–desorption isotherm of the H-G9 clearly indicated a typical of Type IV with H4 type hysteresis loop, which suggested that H-G9 was a mixture of microporous and mesoporous materials [36]. The pore characteristics of H-G9 was more favorable for MB attraction than other glucose-derived hydrochar. This result was in agreement with Figure 3b.

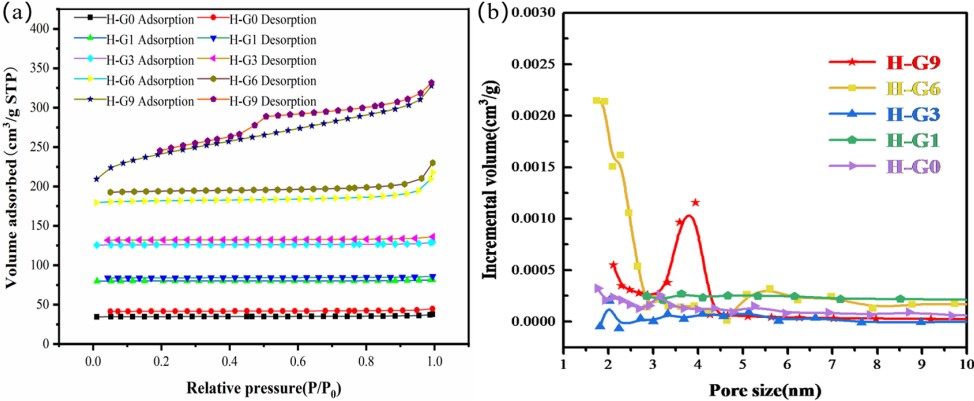

**Figure 3.** (**a**) Nitrogen adsorption/desorption isotherms of hydrochars, (**b**) pore analysis of hydrochars.

Surface area, pore volume, and pore size analysis of hydrochars were estimated by the BET equation and the BJH model (listed in Table 2). The hydrochar obtained at a dosage of 1 mL SA (H-G1) showed a slight decrease in surface area as compared to the hydrochar obtained without SA (H-G0), which was due to the lack of smaller spheres produced in second batch. Observed from H-G1 to H-G9, the $S_{BET}$ and $V_P$ increased in different degrees with an increase in SA. It is worth noting that the pore size distribution of H-G6 was in the range of 5–8 nm (Figure 3b), whereas the $V_P$ was also smaller, probably due to pores being plugged by tarry substances [37,38]. The value of $V_P$ was significantly improved when the SA dosage was increased to 9 mL, indicating that the porous structure and high porosity could be obtained under this acid condition. The largest surface area (296.71 m$^2$/g) occurred in H-G9 and the pore size distribution showed maximum concentration in around 4 nm.

**Table 2.** Structural parameters of the hydrochars.

| Sample | $S_{BET}$ (m$^2$ g$^{-1}$) [a] | $V_P$ (cm$^3$ g$^{-1}$) [b] | $D_P$ (nm) [c] |
|:---:|:---:|:---:|:---:|
| H-G0 | 2.01 | 0.002899 | 12.66 |
| H-G1 | 0.94 | 0.001464 | 14.86 |
| H-G3 | 2.45 | 0.003334 | 14.75 |
| H-G6 | 19.49 | 0.028130 | 18.38 |
| H-G9 | 296.71 | 0.237274 | 4.63 |

[a] $S_{BET}$ is BET surface area. [b] $V_P$ is pore volume. [c] $D_P$ is average pore size.

### 3.4. FTIR Analysis

The hydrochars obtained with and without SA were further analyzed by FTIR to understand the presence of functional groups. As revealed in Figure 4, the broad bands at around 3700–3200 cm$^{-1}$ corresponded to the O-H (-OH, -COOH) bending vibrations, indicating the presence of -OH and -COOH groups on the hydrochars surface [39]. The peaks from 2800 to 3000 cm$^{-1}$ indicated the stretching vibration of the C-H in alkanes or carbonyl [40]. The absorption peaks at 1700 and 1620 cm$^{-1}$ were attributed to C=O (e.g., carbonyl, quinone) and C=C. When the SA dosage was 9 mL (Glu: SA = 1:4), these peaks were significantly strengthened, suggesting a higher degree of carboxylic groups and aromatization in H-G9 compared to other products [41,42]. The peak that appeared at 1180 cm$^{-1}$ could be ascribed to C-O stretching, which strengthened with the increasing dosage of SA. Larger numbers of sulfur-containing functional groups occurred on the surface of the hydrochars when more SA was added. The bands at 1370 cm$^{-1}$, 1030 cm$^{-1}$ and 680 cm$^{-1}$ were related to undissociated SO$_3$H groups, sulfonic acid group (S=O) and S-O group, respectively [43], which possibly played a certain role of the followed adsorption ability for contaminants in the water.

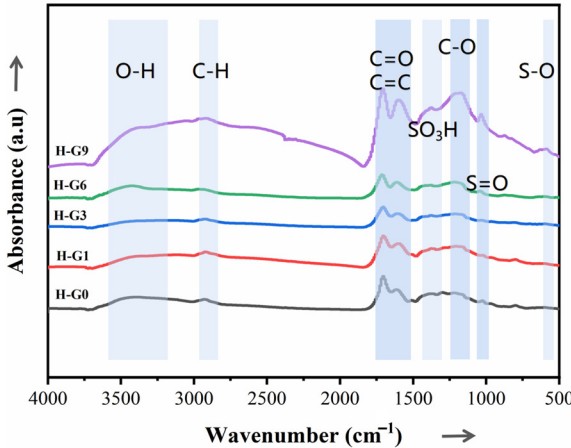

**Figure 4.** FTIR spectra of AC, H-G0, H-G1, H-G3, H-G6 and H-G9.

### 3.5. Zeta Potential

In order to understand the surface charge properties of H-G9, zeta potential measurements were performed in Figure 5, showing that the zeta potential of H-G9 and AC decreased with increasing pH. Overall, the zeta potential value of H-G9 was more negative than AC, the pH of the point of zero charge (pH$_{PZC}$) for AC was 3.4 and the pH$_{PZC}$ for H-G9 was lower, indicating that the surface of H-G9 had much more acid functionalities than that of AC in the same environment. It also could be demonstrated by the infrared data of H-G9, in which many carboxyl and hydroxyl ions were present on the surface. In fact, the hydrochar treated by oxidizing acids had more acid surface functions [44,45]. This indicated the feasibility of H-G9 as adsorbent for contaminant with positive charge surface from solution.

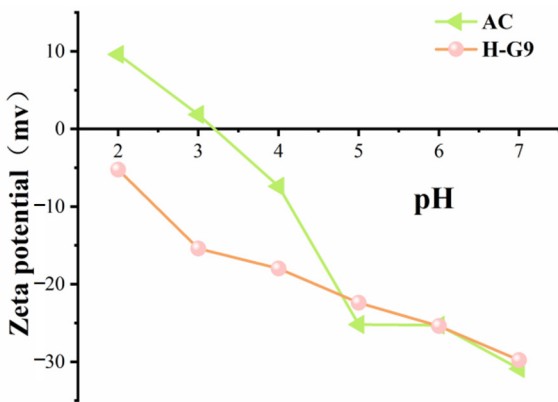

**Figure 5.** Zeta potential analysis of H-G9 and AC.

### 3.6. MB Adsorption Isotherms

3.6.1. Adsorption Kinetics Study

Adsorption kinetics for MB onto the prepared hydrochars was shown in Figure 6a. The curve tended to balance after around 180 min and it was considered that the adsorption equilibrium was reached. Overall, the adsorption capacity of the hydrochars were improved with the increase in acid dosages, and it increased sharply as the dosage of SA increased from 6 to 9 mL. Notably, when the acid was further added, the adsorption performance of H-G12 and H-G15 was not significantly improved. Taking into account the dosage of SA and adsorption performance, the optimal dosage of 9 mL was determined. According to the analysis of FTIR and BET results, the numbers of functional groups and the specific surface area of H-G9 were the largest, which had a great influence on the adsorption capacity.

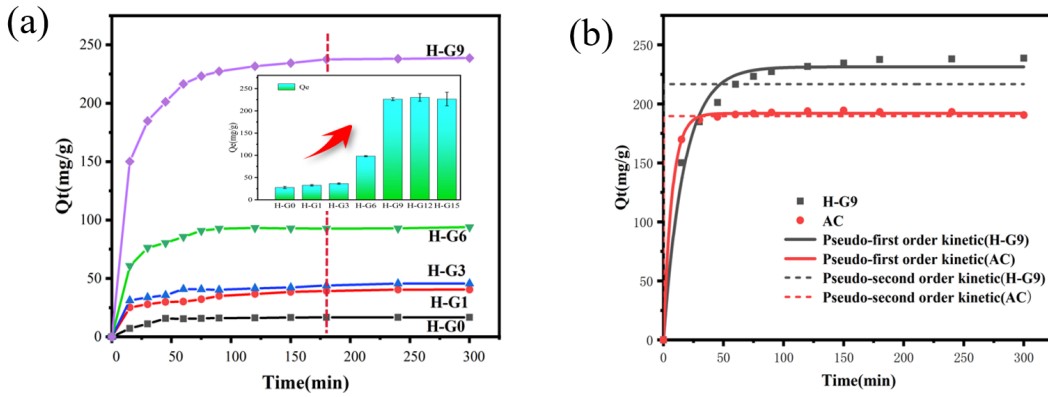

**Figure 6.** (**a**) Adsorption time on $Q_t$ of H-G0, H-G1, H-G3, H-G6 and H-G9, (**b**) Kinetic models of 10 mg/L MB adsorption on H-G9 and AC [$C_0$ = 10 mg/L, Adsorbent Dosage = 15 mg/400 mL, Temp = 25 °C].

The plots of kinetic models of H-G9 and AC for MB were shown in Figure 6b. The fitting parameters of the two models are shown in Table 3, which shows a large difference between the experimental and calculated adsorption capacity, indicating a poor pseudo-second order fit to the experimental data, the pseudo-first order rate model was more suitable for H-G9 ($R^2$ = 0.9863) and AC adsorption ($R^2$ = 0.9991) [46]. Figure 6b shows that adsorption equilibrium of H-G9 was achieved after approximately 180 min with a maximum capacity of 233 mg/g for 10 mg/L initial dye concentration, and the time of the adsorption equilibrium for AC (20 min) was shorter than that of H-G9; however, the adsorption capacity ($Q_e$) of H-G9 was greater.

**Table 3.** Kinetics parameters for MB adsorption onto H-G9 and AC.

| Adsorbent | Pseudo-First Order Kinetic Model | | | Pseudo-Second Order Kinetic Model | | |
|---|---|---|---|---|---|---|
| | $k_1$ (min$^{-1}$) | $Q_e$ (mg g$^{-1}$) | $R^2$ | $k_2$ (g mg$^{-1}$ min$^{-1}$) | $Q_e$ (mg g$^{-1}$) | $R^2$ |
| H-G9 | 0.0582 | 231.38 | 0.9863 | 5.6882 | 216.71 | 0.8471 |
| AC | 0.1419 | 191.92 | 0.9991 | 1.5711 | 189.58 | 0.9856 |

### 3.6.2. Adsorption Equilibrium Isotherms Study

Figure 7a shows the fits of Langmuir and Freundlich models, which were used to understand the interaction between adsorbents and MB molecule in the equilibrium state.

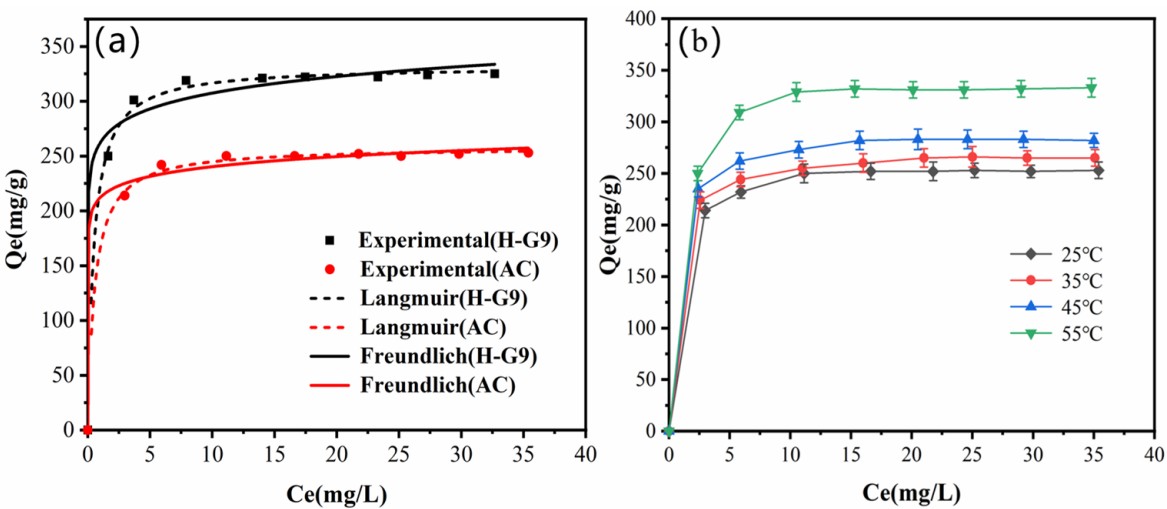

**Figure 7.** (**a**) Adsorption isotherms of MB on the H-G9 and AC, (**b**) adsorption capacity of H-G9 for MB at different temperatures.

Langmuir is a theoretical model which describes behavior of monolayer homogeneous adsorption, and Freundlich isotherm is an empirical equation which describes the formation of multilayers by adsorbate molecules on the heterogeneous surface [47]. By comparing the value of correlation coefficients ($R^2$), the adsorption process of MB onto H-G9 and AC were better described by Freundlich isotherm.

$k_F$ and n of Freundlich model representing the adsorption capacity and adsorption intensity of the adsorbents, respectively. The value of n of H-G9 listed in Table 4 was 5.5279, which was in the range of 1–10, indicating that MB adsorption onto H-G9 was favorable [48]. This was also demonstrated by the separation factor ($R_L$) of Langmuir model [49]. The maximum monolayer adsorption capacity ($Q_m$) was 332.46 mg/g for H-G9. Table 5 presented the maximum adsorption capacities of different adsorbents for MB from the previous literature and this study. Obviously, the H-G9 was an effective adsorbent to remove MB from aqueous solutions compared with many other adsorbents, and the value of $Q_m$ was much higher than others.

**Table 4.** Adsorption isotherms for MB adsorption onto H-G9 and AC at pH 6.0 and 25 °C.

| Adsorbent | Langmuir Isotherm Model | | | Freundlich Isotherm Model | | |
|---|---|---|---|---|---|---|
| | $Q_m$ (mg g$^{-1}$) | $K_L$ (L mg$^{-1}$) | $R^2$ | $K_F$ | n | $R^2$ |
| H-G9 | 332.46 | 2.2146 | 0.9682 | 267.31 | 5.5279 | 0.9880 |
| AC | 259.37 | 1.8172 | 0.9379 | 215.56 | 9.7628 | 0.9944 |

**Table 5.** The maximum adsorption capacity ($Q_m$) of different adsorbents for MB.

| Adsorbent | $Q_m$ (mg g$^{-1}$) | References |
|---|---|---|
| Lignin derived carbon-based | 234.65 | Liu et al. (2021) [50] |
| Glucose-based hydrochar | 179.8 | Pak et al. (2022) [51] |
| Hydrochar from Pomegranate Peels | 194.9 | Hessien et al. (2022) [52] |
| The carbon sample prepared with FeCl3 and KOH | 248.9 | Grassi et al. (2022) [53] |
| Nano-carbon material (N-CM) adsorbent | 120.77 | Liang et al. (2022) [54] |
| The chitosan-glutaraldehyde/activated carbon | 300.75 | Cao et al. (2022) [55] |
| Glucose-based porous carbon material | 332.46 | The present study |

The adsorption isotherms of MB on H-G9 at 25, 35, 45 and 55 °C are shown in Figure 7b. In the case of increasing temperature, the amount of adsorbed dye increased, indicating that MB adsorption onto H-G9 may be endothermic. The adsorption driving force was affected by the reaction temperature, and higher temperature favored the MB removal over.

### 3.7. Effect of Solution pH

pH is an vital factor that affects the adsorbate onto adsorbent. As shown in Figure 8, the extraction efficiency for MB increased with an increase in pH from 2 to 10. When the pH was increased (2–10), the adsorption performance increased. As shown in Figure 5, the zeta potential offered the possible mechanism about the electrostatic interaction between adsorbent and adsorbate. The $pH_{PZC}$ of H-G9 decreased with increasing pH, and the more active sites on H-G9 surface, the more it can adsorb cationic dye (MB).

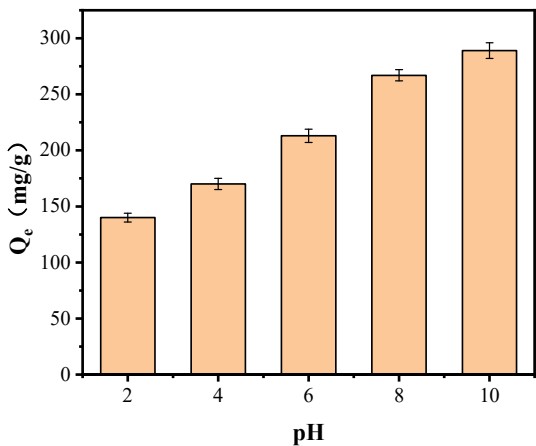

**Figure 8.** Influence of pH on the adsorption of MB by H-G9.

### 3.8. Adsorbent Regeneration

The recovering ability of adsorbents determined its commercial application value. A new HTC regeneration method was investigated in this study, which did not require drying of the saturated adsorbent. The adsorbent was repeated three times for experiments and the regeneration efficiency of the H-G9 were 88%, 80% and 70%, respectively. Furthermore, methods reported in the literature were also used to recovered H-G9 by using ethanol [36,50] and HCl [56] and showed that the regeneration efficiency of the first time only reached 80% and 44%. Obviously, these results indicated that HTC regeneration method was better, and the conditions need to be further optimized in the future.

### 4. Conclusions

In this study, porous carbon material was successfully prepared by the simple one-step hydrothermal carbonization of glucose in the presence of SA. The addition of SA plays a vital role in chemical properties and structure of products. When the SA dosage was increased to 9 mL, the BET and $V_P$ of sample (H-G9) were developed significantly.

The micrograph of H-G9 showed that large aggregates with a porous structure appeared, which were accumulated by nanoparticles. The results of FT-IR indicated that a plenty of sulfur-containing and oxygen-containing functional groups occurred on the surface of the hydrochar. Zeta potential analysis indicating that the surface of H-G9 had much more acid functionalities than that of AC. H-G9 was applied as a cost-effective adsorbent for attracting cationic MB dye with a maximum monolayer adsorption capacity of 332.46 mg/g. Additionally, hydrothermal regeneration experiments proved that H-G9 has good reproducibility. These encouraging results demonstrated that the porous carbon material is highly promising for removal of MB from wastewater, and this study is to provide technical support for the one-step preparation of porous carbon materials from biomass.

**Author Contributions:** Z.Y.: overall data collection, analysis of data, and writing—original draft; W.Z.: funding acquisition, resources, modification of manuscript in terms of language and decided the way of representation; X.Y.: methodology design, conceptualization and supervision. All authors have read and agreed to the published version of the manuscript.

**Funding:** This study was financially supported by China State Construction Innovation Project (CSCEC-2012-Z-14).

**Data Availability Statement:** All data generated or analyzed during this study are included in this article.

**Conflicts of Interest:** The authors declare no conflict of interest.

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
