# Peer review of "Fabricating Porous Carbon Materials by One-Step Hydrothermal Carbonization of Glucose"

_processes, doi:10.3390/pr11071923_

Round 1

Reviewer 1 Report (New Reviewer)

Fabricating Porous Carbon Materials by One-Step Hydrothermal Carbonization of Glucose has been reviewed and the following observations have been made:

1.      Author should add the thermal gravimetric analysis (TGA) of formulated porous carbon materials to check their stability.

2.        XRD analysis of the material is missing.

3.      Conclusion section is very short, it should be more informative.

4.      10 ppm of Methylene blue (MB) concentration is very low, experiments must be performed with a higher concentration of MB.

5.      Graph related to the percentage removal of MB with time can be added with respect to the adsorbent dose of the material used in the experiment.

6.      Authors should mention the materials' futuristic use and stability on a commercial level.

7.      Typographical errors should be removed.

8.      In a scientific report, please try to avoid using words like I, we, our, you, and they. We are presenting a scientific thing but not an oral presentation which can be extended with exaggeration.

The grammatical errors should be removed.

Author Response

Reviewer 2 Report (New Reviewer)

This manuscript is focused on the production of porous carbon materials by acid- catalyzed hydrothermal carbonization of glucose. The effect of sulfuric acid dose on the structure, the morphology and the presence of functional groups on the surface of the carbonaceous material is analyzed and optimized for methylene blue adsorption. The regeneration of the material is also studied.

The manuscript is well structured and shows interesting scientific results. However, given the large amount of biowaste generated and the wide variety of pollutants in wastewater, the use of some of these instead of model compounds (glucose and MB) would be more desirable for this study, finding an application to this type of waste of environmental concern.

Specific comments:

-          Sulfuric acid was added in the HTC system, was the possibility of using other acids (hydrochloric acid, nitric acid …) not considered to study the yield and chemical properties of the hydrochar?

-          Lines 60-62 “this study concerns the production of glucose-based porous carbonaceous material by a one-step process in the presence of acid, and will provide a new route for the future conversion of biomass”. However, there are already several works in the literature on this matter using real waste.

-          Line 108 “M indicates quality…”, mass or amount.

-          Adsorbent regeneration experiment section. Why do you change the concentrations of HC and MB in the regeneration experiment with respect to the initial ones used in the adsorption test?

-          Line 217 “This result was in agreement with the Fig. 3b.36” Why do you use a reference in a figure of yours?

-          Adsorption kinetics study need mor discussion. The values of the parameters should be compared with those found in other works in the literature.

-          The legend in figure 3a is not clearly visible.

-          Please, use the same colors for HG9 (black) and AC (red) in figures 6b and 7a.

Round 2

Reviewer 1 Report (New Reviewer)

The required changes in the manuscript have been made by the authors. Now it can be accepted for publication.

Reviewer 2 Report (New Reviewer)

I acknowledge the answers of authors to all my comments and the clarifications and changes on the manuscript.

The revised version of the manuscript can be accepted for publication. 

This manuscript is a resubmission of an earlier submission. The following is a list of the peer review reports and author responses from that submission.

Round 1

Reviewer 1 Report

The work focuses on the preparation of carbon materials from glucose by acidic hydrothermal carbonization (HTC). Authors use sulphuric acid to produce porosity in the carbonized material. The manuscript needs substantial improvement before being published. This topic has been previously studied and the work does not deepen the knowledge of these materials and their application as adsorbents. In my opinion, the work is not of high enough quality to be published in Processes. Much more work needs to be done before resubmitting it for review.

General comments

Experimentally the work is very simple. The authors only study the addition of several doses of sulphuric acid to an initial mass of glucose and then it is carbonized at 180ºC and 4 h. There is no justification of the acid doses (which should be expressed as mass ratio with respect to the starting glucose and not as volume of acid), and the operating conditions used.

The authors should establish other operating conditions (especially temperature) to improve the results and optimize the process. Using fixed conditions does not allow conclusions to be drawn. In fact, in the carbonization stage, the experiments are carried out only once using several reactors that are induced in a furnace at 180ºC (one experiment). The work mainly focuses on the characterization of the carbonized materials by different techniques (elemental analysis, SEM, BET area, Fourier transform infrared and point of zero charge).

In the adsorption stage, again, the operating conditions are fixed, which does not allow the process to be optimized. It would be necessary to carry out further adsorption studies at other temperatures and pH, which would allow to know in depth the adsorption process and to carry out a thermodynamic study.

The manuscript must be proofread by an English speaker.

The manuscript does not include an adequate indication of the reproducibility of the results, level of experimental error and the statistical significance of the results.

Throughout the text, numbers must be separated from units.

Keywords

Replace “porous carbonaceous materia” by “porous carbon materials”

Abstract

The abstract should be rewritten for a better understanding of the work by the readers.

Lines 8-15 are unnecessary. The summary should include experimental results. Characterization studies are included in the materials and methods section. The reader should be aware of the most interesting results in this section.

The way of expressing the amount of acid used (9 mL) is improper. With respect to what amount of glucose? Giving a specific value for the volume used does not provide information. Glucose:acid mass ratios should be used, both in the abstract and in the rest of the manuscript.

The BET area cannot be given with decimals. The error of the method does not allow that precision.

Line 16-17. “… higher abundance of functional groups on the surface than that of other samples …”. What functional groups? How do these functional groups affect the adsorption capacity?

Line 17-18. ex-18 “… excellent adsorption efficiency for methylene blue… ”. With respect to what materials? The adsorption of methylene blue has been extensively studied in the literature. These statements cannot be made, since the adsorption of this compound with carbon materials is much higher than that reported by the authors.

Lines 23-24. The statement: “This indicated the extraordinary potential of H-G9 as adsorbent for aqueous contaminant removal” is not true. The authors can only state that they observe adequate uptake for methylene blue retention, in no case extendable to other pollutants in the aqueous phase. They could test pollutants with different pKa to establish such a claim, and certainly the work would be of greater interest.

Introduction

The introduction is very poor and poorly written. The authors should discuss information from the literature of carbon materials obtained by hydrothermal carbonization and their subsequent use as adsorbents. There are numerous papers they can use, many of them published in MDPI journals (Applied Sciences, Molecules, Resources, etc.).

Authors should change the way they write and focus on discussing information rather than citing articles one by one.

Materials and methods

The title of this section should be corrected: “Materials and methods” instead of “Material and methods”.

Comments have been made regarding the experimental procedure. A single carbonization experiment in a furnace, without repetitions and, therefore, without statistical validation. The same can be said for the adsorption experiments.

The denominator of equation 1 is not understood. It should appear: “Mass of dry glucose” instead of “Mass of dry feed glucose”.

The Langmuir isotherm equation should be modified to express the adsorption capacity as a function of the methylene blue concentration at equilibrium.

Results and Discussion

The results of hydrochar yield and organic matter in the liquid phase are quite strange. In my opinion I think they are changed. In all the articles reported in the literature, the addition of acid in hydrothermal carbonization favors hydrolysis reactions and, therefore, reduces the yield to solid, favoring the solubilization of organic matter. I believe that the authors have not studied the literature in depth.

The discussion of the results is very poor. One simple example will suffice (lines 140-141): “When the dosage of SA increased from 6 to 9ml, the COD of process water continued to decrease from 1440mg/L to 512mg/L”. It is hard to believe that this is happening.

Table 1. Elemental composition results do not close the balance sheet.

Table 2 shows values of BET area (m2/g) and particle diameter (nm) to 4 decimal places. This is non-scientific. It is also unscientific to show BET area values of 0.9389 or 2.0070, which are below the error incurred by the technique.

The FTIR results are qualitative and do not provide much relevant information about the functional groups of the materials. Many prestigious journals do not allow the use of FTIR to report information on surface functional groups.

It is not surprising that H-G9 material, that obtained with the highest dose of sulfuric acid, is the only one that reports a significant methylene blue adsorption capacity, since it is the only material that has developed some BET area.

The authors do not explain the adsorption mechanism, but only report information on the experiments performed.

The authors use an activated carbon for comparative purposes, but its origin and characteristics are unknown. Only functional groups and zeta potential.

Conclusions

The conclusions must be rewritten. As the study is formulated, it is very difficult to draw conclusions. In any case, they cannot include sentences identical or similar to those in the abstract.

The manuscript must be proofread by an English speaker.

Reviewer 2 Report

The glucose-based porous carbonaceous materials were obtained by the addition of sulfuric acid with different dosages in the HTC. The materials were characterized by SEM, BET, FTIR, and EA and used as adsorption materials to remove MB. The adsorption results were discussed in detail. Therefore, the manuscript can be considered for publication after addressing the following issues.

1.      Authors need to check the full paper carefully. Some terms and phrases need to be corrected.

For example, “Fig.” was used in the text of the article, however, each graph is labeled with the full name (Figure), which needs to be consistent.

The format of the references needs to be examined, especially some journal names, which may be abbreviated or full.

Line 185, 226, 239, 281, the format of each title needs to be uniform.

Line 98, Please correct the following sentence, “Where C0, Ct and Ce representative the initial, at time t and equilibrium concentrations of MB solution (mg/L), respectively.”

Line 273, double punctuation at the end of sentences. “The maximum adsorption capacities (Qm) was 332.46 mg/g for H-G9,.”

Line 280, the format of Table 5 needs to be adjusted so that the carbon material corresponds to the adsorption capacity.

2.      Line 137, I looked forward to knowing how to obtain this conclusion, I think it needs to add some explanation or reference to support this sentence. “This might be due to the degree of aromatization is improved.”

3.      Line 77, The analytical process for COD needs to add some content or reference appropriately.

4.      By comparison the value of correlation coefficients (R2), the adsorption process of MB onto H-G9 and AC were better described by Freundlich isotherm. Therefore, the result of the maximum adsorption capacity (332.46mg/g) simulated by the Langmuir model is not appropriate.

Reviewer 3 Report

Dear Authors, sorry but I cannot recommend this work for publication.

Reasons: lack of novelty and huge mistakes:

let me summarize, glucose à hydrothermal treatment (it was many, many times investigated) à acid addition (it was done e.g. 10.1016/j.jcis.2010.11.072) à methylene blue adsorption (because what else? it was also, investigated many times; you can say your results are the best – Tab. 5, but why you mentioned only the materials with lower ads. Cap.? Look e.g. for 10.1021/acs.jpcb.1c08294 there is ca 2500 mg/g) kinetic and Langmuir model used, do not bring us any novelty.

So, based on the fact that glucose-derived HTCs are known and rehearsed from every side, I cannot agree for this text publication.

More:

-          Experimental section needs increasing; especially section 2.3 Characterisation

-          Please, do not use the baseline in FTIR analysis!!

-          How did you measure the zeta potential? Your carbons agglomerate to ca. 10-20 um. (i) How is it possible to measure their ZP? (ii) how is it possible to form a stable solution (ca. -30 mV) from these materials?

BTW, I’d rather worry, as Figs 4 & 5 suggest that huge amount of H2SO4 is still inside the material

-          Fig. 3 – add the scale; differentiate adsorption and desorption branches

-          Tab 4. The parameters for Freundlich Ads Isoth. are wrong, (I didn’t check the others)

-          I’m afraid that you do not understand the hydrothermal process mechanism; the yield you calculated is incorrect, and say nothing.

BTW, what you mean “chemical oxygen demand (COD) were performed according to standard methods.”? I do not understand “the COD of process water continued to decrease from 1440mg/L to 512mg/L.” Did you measure water?

-          The materials you obtained are rather microporous so BJH for PSD is incorrect

-          Why do you show BET with 0.0001 accuracy? This is absolutely wrong!!!!

-          The same with other parameters

-          I cannot agree with the statement: “The adsorption capacity was as high as 332.46 mg/g for MB, which was well beyond that of commercial activated carbon.

-          It also does not mean that “This indicated the extraordinary potential of H-G9 as adsorbent for aqueous contaminant removal.

English is poor

Round 2

Reviewer 1 Report

The manuscript is not of enough quality to be published in Processes or any other journal with an impact index. It is poorly written and based on previous results of other authors that are not even cited.

Extensive editing of English language required.

Reviewer 3 Report

-----